# Identification of Anther Length QTL and Construction of Chromosome Segment Substitution Lines of *Oryza longistaminata*

**DOI:** 10.3390/plants8100388

**Published:** 2019-09-29

**Authors:** Takayuki Ogami, Hideshi Yasui, Atsushi Yoshimura, Yoshiyuki Yamagata

**Affiliations:** Plant Breeding Laboratory, Faculty of Agriculture, Kyushu University. 744, Motooka, Nishi-ku, Fukuoka 819-0395, Japan; ohgammy9.30@gmail.com (T.O.); hyasui@agr.kyushu-u.ac.jp (H.Y.); ayoshi@agr.kyushu-u.ac.jp (A.Y.)

**Keywords:** anther length, cell elongation, genetic architecture, outcrossing, perennial species, rice

## Abstract

Life histories and breeding systems strongly affect the genetic diversity of seed plants, but the genetic architectures that promote outcrossing in *Oryza longistaminata*, a perennial wild species in Africa, are not understood. We conducted a genetic analysis of the anther length of *O. longistaminata* accession W1508 using advanced backcross quantitative trait locus (QTL) analysis and chromosomal segment substitution lines (CSSLs) in the genetic background of *O. sativa* Taichung 65 (T65), with simple sequence repeat markers. QTL analysis of the BC_3_F_1_ population (n = 100) revealed that four main QTL regions on chromosomes 3, 5, and 6 were associated to anther length. We selected a minimum set of BC_3_F_2_ plants for the development of CSSLs to cover as much of the W1508 genome as possible. The additional minor QTLs were suggested in the regional QTL analysis, using 21 to 24 plants in each of the selected BC_3_F_2_ population. The main QTLs found on chromosomes 3, 5, and 6 were validated and designated *qATL3*, *qATL5*, *qATL6.1*, and *qATL6.2*, as novel QTLs identified in *O. longistaminata* in the mapping populations of 94, 88, 70, and 95 BC_3_F_4_ plants. *qATL3*, *qATL5*, and *qATL6.1* likely contributed to anther length by cell elongation, whereas *qATL6.2* likely contributed by cell multiplication. The QTLs were confirmed again in an evaluation of the W1508ILs. In several chromosome segment substitution lines without the four validated QTLs, the anthers were also longer than those of T65, suggesting that other QTLs also increase anther length in W1508. The cloning and diversity analyses of genes conferring anther length QTLs promotes utilization of the genetic resources of wild species, and the understanding of haplotype evolution on the differentiation of annuality and perenniality in the genus *Oryza*.

## 1. Introduction

Life histories and breeding systems strongly affect the genetic diversity of seed plants. Annuals tend to allocate their resources to sexual reproduction to produce as many flowers as possible for the one-time dispersal of seeds. By contrast, perennial species tend to primarily allocate resources to vegetative growth because of the need to occupy physical space, and to use local water and nutrient resources over an extended life span depending on the ecological circumstances [1].

Perennial species tend to show higher heterozygosity when compared with the annual species or domesticated species [2]. Heterozygosity has been found to correlate with fitness-related traits, such as survival probability, reproductive success, and disease resistance [3,4,5,6,7]. Populations of perennials maintain high genetic diversity by producing few but large floral organs, and by promoting relatively high outcrossing rates with mechanisms to prevent self-pollination—such as self-incompatibility or monoecious flowers [8,9]. An understanding of the genetic architecture of a favorable trait that promotes outcrossing will help to answer a question: What set of polymorphisms or genes contribute to outcrossing characteristics? However, the genetic basis of breeding system-associated traits is not fully understood.

There are eight species of the genus *Oryza* with the AA genome: Two Asian wild species (*O. rufipogon* Griff. and *O. nivara* Sharma et Shastry), two wild species in Africa (*O. longistaminata* A. Chev. & Roehr. and *O. barthii* A. Chev.), one wild species in South America (*O. glumaepatula* Steud.), one wild species in Australia (*O. meridionalis* Ng.), and two cultivated species [10]. Of the two cultivated rice species, *O. sativa* L. (Asian rice) is domesticated from the Asian perennial wild species *O. rufipogon*, and *O. glaberrima* Steud. (African rice) is domesticated from the African annual wild species *O. barthii*. The Asian cultivated and wild species form a species complex in which the production of hybrid progeny is possible. The differentiation of annual (*O. nivara*) and perennial (*O. rufipogon*) species reflects adaptation to their ecological niches [11,12]. In Africa, the annual species *O. barthii* and the perennial species *O. longistaminata* occupy different ecological habitats. *Oryza longistaminata* is characterized by a rhizome [13], and has a particularly large anther [14], as indicated by the species name ‘*longistaminata*’ (long stamen). Thus, the anther is one of the key traits of this species. Additionally, it has high heterozygosity [15] that is likely due to its self-incompatibility and high outcrossing rate, achieved by the large reproductive organs [16]. Cultivated rice is a self-pollinated species with an outcrossing rate of less than 4% [17]. The outcrossing rate of perennial wild species tends to be higher than that of cultivated species. That of *O. longistaminata* and *O. rufipogon* reaches 100% in certain combinations of hybridization, although the rate for wild species typically ranges from 3.2% to 50% [18,19]. To fully understand perenniality and their outcrossing characteristics, we need to investigate the genetic architectures of their reproductive systems.

Chromosomal segment substitution lines (CSSLs) facilitate the genetic analysis and characterization of quantitative traits of donor varieties, or species in the genetic background of a recurrent parent. CSSLs are powerful genetic tools for the identification of quantitative trait loci (QTLs) [20,21,22,23] in trials across different years and environments [24,25,26]. Studies of anther QTLs have been conducted with F_2_ populations and recombinant inbred line (RIL) populations derived from the interspecific crosses between *O. rufipogon* and *O. sativa* [27,28,29], or intraspecific crosses between the *indica* and *japonica* subspecies of *O. sativa* [30]. However, QTLs that confer anther length in *O. longistaminata* have not been examined and validated using near-isogenic lines (NIL) in the genetic background of *O. sativa*.

In this study, we elucidated the genetic basis of anther length in *O. longistaminata*, by using advanced backcross QTL analysis and CSSLs in the genetic background of *O. sativa* L. cv. Taichung 65 (T65). QTLs conferring anther length identified in BC_3_F_1_ and BC_3_F_2_ populations were validated in the BC_3_F_4_ population, in which single QTL regions were segregating. We validated the QTLs again by constructing CSSLs of *O. longistaminata*. NILs were evaluated to characterize the histological cause of increased anther length. CSSLs chromosome segments derived from *O. longistaminata* acc. W1508 were named as W1508ILs, using the term ‘introgression lines’ (ILs) to refer to CSSLs based on interspecific hybridization.

## 2. Results

### 2.1. Genetic Variation in Anther Length in the Backcrossed Population

We measured the anther length of *O. barthii*, *O. glumaepatula*, and *O. longistaminata* (Figure 1) to evaluate the genetic variation in anther length. The anther length of *O. longistaminata* were longer than those of all *O. barthii* and *O. glumaepatula* accessions, except for *O. glumaepatula* W1183. To identify the QTLs derived from *O. longistaminata* in a uniform genetic background, we developed CSSLs carrying W1508 (*O. longistaminata*) chromosomal segments in the genetic background of *O. sativa* T65.

We developed 372 BC_3_F_1_ plants by recurrent backcrossing to F_1_, BC_1_, and BC_2_ plants to develop the CSSLs of W1508 (Figure 2). We genotyped 100 BC_3_F_1_ plants, and selected 26 BC_3_F_1_ plants that covered as much of the W1508 genome as possible, to minimize the number of candidate CSSLs (Appendix A). Simple interval mapping (n = 100) suggested that *O. longistamina* alleles at one QTL on chromosome (Chr.) 3, one QTL on Chr. 5, and two QTLs on Chr. 6 increased the anther length (Appendix A). The other peaks were below the experiment-wise threshold levels of the logarithm of odds (LOD) at LOD_α = 0.05_ = 2.65 and LOD_α = 0.01_ = 3.40. Multiple QTL analysis using forward/backward model selection also suggested that these four QTLs additively increase anther length without epistasis, which explains more than half (55.5%) of the phenotypic variation in the BC_3_F_1_ population (Appendix A, Appendix A).

To construct CSSLs, and to perform regional QTL analysis using 21 to 24 plants of the BC_3_F_2_ populations derived from the selected 26 BC_3_F_1_ plants covering the whole genome, we defined a heterozygous “target region” as a genomic region in BC_3_F_1_ plants for the fixation of the W1508-derived segments on the homozygous condition in the progeny (Appendix A). The anther length of T65 was 2.33 mm and that of W1508 was 4.33 mm. The mean anther length of the 26 BC_3_F_2_ lines ranged from 2.23 to 2.85 mm. Nine BC_3_F_2_ lines had a significantly higher mean anther length than T65, but none had a significantly lower mean anther length (Figure 3). Our working hypothesis was that the W1508 segments in the 9 BC_3_F_2_ lines with a higher mean anther length (boxed in Appendix A) most likely had QTLs associated with longer anther length. Simple interval mapping in the 26 BC_3_F_2_ populations at the target regions detected 13 minor QTLs on Chrs. 1, 2, 3, 4, 5, 6, 9, 10, and 11 (Appendix A).

### 2.2. Validation of QTL

For the identification of major QTLs conferring anther length, we conducted QTL analysis on BC_3_F_2_ 13, 16, and 21 families that had the highest anther length means of 2.88 mm, 2.82 mm, and 2.68 mm, respectively (Figure 4). In the BC_3_F_2_ 21 population (n = 44), there was segregation on chromosomes 3, 6, and 11 (Figure 4a). The QTL positively regulating anther length located between *RM3436* and *RM5959* on chromosome 3 was named *qATL3.3*. It was validated using the more advanced BC_3_F_4_ 6 (n = 96) generation at 151.5 cM on chromosome 3, with the LOD value 18.27. *qATL3.3* explained 58.4% of the phenotypic variance in the BC_3_F_4_ population, with 0.17 mm of additive effect and 0.03 mm of dominant effect (Appendix A).

Among the 71 plants in the BC_3_F_2_ 13 population, the range of anther length was 2.26–3.32 mm, with a population mean of 2.82 mm. Simple interval mapping revealed two QTL: *qATL3.1* at 137.6 cM on Chr. 3, and *qATL5* at 122 cM on Chr. 5. Segregation distortion at *qATL5* was detected at the closest marker, *RM1054*, with a ratio of 22 T65 homozygotes: 43 T65 heterozygotes: 6 W1508 homozygotes. *qATL3.1* and *qATL5* were each independently validated in the more advanced BC_3_F_4_ 3 and nine populations, respectively, derived from the BC_3_F_2_ 13 population (Figure 4b,c). *qATL3.1* was detected again at 142.5 cM on Chr. 3, with an LOD score of 15.70, an additive effect of 0.19 mm, and a dominance effect of −0.002 mm, explaining 47.6% of the phenotypic variance (Appendix A). *qATL3.3* and *qATL3.1* seemed to be identical QTLs, because substitution of the *O. longistaminata* chromosome segment ranged between the SSR markers *RM3525* and *RM5959*. We unified the QTL names *qATL3.3* and *qATL3.1* as the single QTL *qATL3.1*. In BC_3_F_4_ 9, *qATL5* was detected again at 121.4 cM on the long arm of Chr. 5, with an LOD score of 17.28, an additive effect of 0.16 mm, and a dominance effect of 0.08 mm, explaining 59.6% of the phenotypic variation.

The BC_3_F_2_ 16 population (n = 86) segregated at genomic regions on Chrs. 1, 2, 3, 6, 8, and 10, with anther lengths ranging from 2.21 to 3.10 mm and a population mean of 2.68 mm (Figure 4d,e). In the QTL analysis, the QTLs derived from *O. longistaminata* on Chr. 6 (*qATL6*) increased anther length. Because the genomic region of *qATL6* ranged widely from 13.8 to 114.8 cM, with LOD scores greater than the 5% significance level, we assumed that multiple QTLs on Chr. 6 were linked (Figure 4f). We examined two progeny lines of the BC_3_F_2_ 16 population, BC_3_F_4_ 17 and 14, that had the separated *O. longistaminata* chromosome segments on the short and long arms, respectively. In BC_3_F_4_ 17, QTL *qATL6.1* was detected at 38.3 cM on Chr. 6S (Figure 4d,f), which, with an additive effect of 0.06 mm and a dominance effect of 0.08 mm, explained 48.8% of the phenotypic variation in anther length. *RM7023* segregation distortion of the nearest-neighbor marker, *RM7023*, gaved at a ratio of 25 T65 homozygotes: 47 T65 heterozygotes: 1 W1508 homozygotes. In BC_3_F_4_ 14, a QTL designated as *qATL6.2* was detected on the long arm of Chr. 6 (Figure 4e,f) at *RM7309*, with an LOD score of 7.91, an additive effect of 0.08 mm, and a dominance effect of 0.04 mm.

### 2.3. Characterization of Anther Length QTL Using Near-Isogenic Lines

To understand the cytological characteristics of the anther length in near-isogenic lines (NILs) for *qATL3.1*, *qATL5*, *qATL6.1*, and *qATL6.2*, we investigated the longitudinal lengths of epidermal cells (cell length, CL) in the mid-anther region. The anther length of the NILs for *qATL3.1*, *qATL5*, *qATL6.1*, and *qATL6.2* were significantly longer than those of T65, according to Dunnett’s multiple comparison (Figure 5a). The CL of the *qATL3.1* NIL was significantly longer than that of T65, but that of the *qATL5* and *qATL6.1* NILs was not different (Figure 5b). The average CL in the *qATL6.2* NIL was significantly smaller than that in T65. These results suggest that the positive effect of *qATL3.1* on anther length was due primarily to cell elongation in the longitudinal direction, whereas that of the other QTLs, particularly *qATL6.2*, might have been due to cell number. Next, we investigated the anther length and pollen grain numbers (PGN) of T65, W1508, and the QTL NILs. PGNs increased in the order of 1910.0 in T65, 2091.1 in the *qATL6B* NIL, 2598.1 in the *qATL6.1* NIL, 2690.1 in the *qATL5* NIL, 3058.8 in the *qATL3.1* NIL, and 6862.2 in W1508.

### 2.4. Evaluation on W1508Ils

The target chromosome segments of W1508 selected in BC_3_F_1_ were fixed to homozygotes for the W1508 genes in BC_3_F_2_, BC_3_F_3_, BC_3_F_4_, and BC_3_F_5_, to construct the W1508ILs (Figure 6). We selected 32 lines covering the whole genomic region of W1508 using 125 SSR markers. Introgression of W1508 chromosomes did not occur at *RM3148* on Chr. 1; *RM5635* on Chr. 4; *RM2381* on Chr. 7; *RM8219* on Chr. 9; *RM7557*, *RM26090*, and *RM5918* on Chr. 11; or *RM3455*, *RM7195*, *RM1261*, and *RM309* on Chr. 12. The mean anther length in four or five plants increased significantly in W1508ILs 1, 2, 3, 7, 8, 9, 15, and 29, and decreased significantly in W1508IL 20 (Appendix A). W1508ILs 8 and 9 carried *qATL3*, and W1508IL 15 carried *qATL5*. However, the mean anther length in W1508IL 17 (carrying *qATL6.1*) or 18 (carrying *qATL6.2*) did not increase significantly in five plant samples, suggesting that four or five replicates were not sufficient to detect the QTLs, despite being major QTL, for increased anther length. The minor QTLs detected in the BC_3_F_2_ populations were likely associated with the mean anther length in W1508ILs 1 and 2 at *RM3604*, and in W1508IL 7 at *RM5755*.

## 3. Discussion

Polygenes or QTLs control most agronomic traits. For genetic analysis of quantitative traits, segregating populations such as F_2_, RIL, and advanced backcross populations allow complicated genetic architectures to be disassembled to a few genetic elements, thereby increasing the power of detection. Because our objective was to construct a platform from which to analyze the genetic difference between *O. sativa* and *O. longistaminata*, we evaluated the phenotypic character anther length, which is a key character in *O. longistaminata,* and genotyped BC_3_ populations and W1508ILs. We used genome-wide QTL exploration and model fitting procedures, and determined that phenotypic variation in anther length was explained by four QTLs on Chrs. 3, 5, and 6. Our subsequent single-factor segregation analyses of QTLs in BC_3_F_4_ populations validated *qATL3.1* (with an additive effect of 0.17 mm), *qATL3.3* (0.19 mm), *qATL5* (0.16 mm), *qATL6.1* (0.06 mm), and *qATL6.2* (0.08 mm). *qATL3.1* and *qATL3.3* were detected near one another on Chr. 3. Although we considered these two QTLs to be identical, their LOD peaks indicated different genomic positions, most likely owing to the slight segregation distortion around *qATL3.3* (df = 2, P = 0.006). QTLs regulating anther length were also reported around the same regions in *O. rufipogon* [27,28], and close to the *qATL5* region in F_2_ populations derived from *O. sativa* × *O. rufipogon* [21], and in RILs derived from *O. sativa* subsp. *indica* × *O. rufipogon* W1944 [28,29]. QTLs that increase anther length have also been detected on Chr. 6 [27,28]. However, these previously identified QTLs have not been validated in *O. sativa* backcross populations in the absence of effects of other QTLs. Therefore, they may not be identical to the QTLs found here.

QTL analysis in NILs showed that the increase in anther length at flowering was likely due to cell elongation under the influence of *qATL3*, *qATL5*, and *qATL6.1*, and to cell division under the influence of *qATL6B*. The development of stamens is divided into two phases [31]. The early phase features the morphogenesis of floral organs, including stamen tissues, from the floral meristem. The late phase comprises pollen maturation, filament elongation, anther dehiscence, and pollen release, which are modulated by auxin (AUX), gibberellin (GA), and jasmonate (JA) [32,33,34,35]. AUX and GA can also regulate the early phase [34,36]. The expression of a MYB transcription factor, *HvGAMYB*, in barley, is upregulated by GA in early anther development in the nuclei of the epidermis, endothecium, middle layer, and tapetum, leading to sterility and shorter anther length [37]. Because AUX, GA, and JA affect stamen morphogenesis and male gametogenesis, the QTLs might be involved in epidermal cells via plant hormone signaling.

Because the mean anther length in T65 was 2.27 mm, we expected a plant homozygous for the *O. longistaminata* alleles at the four QTLs without epistatic effect among them to have an anther length of 3.21 mm (2.27 + 2 (0.17 + 0.16 + 0.06 + 0.08) mm), due to the additive effects of *qATL3.1*, *qATL5*, *qATL6.1*, and *qATL6.2,* or, of 3.25 mm (2.27 + 2 (0.19 + 0.16 + 0.06 + 0.08) mm) due to *qATL3.3*, *qATL5*, *qATL6.1*, and *qATL6.2*. As the anther length of W1508 was longer at 4.33 mm, additional, unidentified QTLs and interactions with minor effects are likely hidden in the *O. longistaminata* genome. QTL exploration in BC_3_F_2_ populations by simple interval mapping using only 21 to 24 plants in each population suggested 10 chromosomal regions with effects that increased anther length, and three that decreased anther length. We expected that the target regions independently segregated with *qATL3.1*, *qATL5*, *qATL6.1*, and *qATL6.2* in the QTL analysis of BC_3_F_2_ generations, and that the QTL analysis nested in each of the BC_3_F_2_ populations eliminated the genetic “noise” caused by *qATL3.1*, *qATL5*, *qATL6.1*, and *qATL6.2* in some of the BC_3_F_2_ populations, by fixation of the homozygous condition for the T65 allele (Appendix A). The BC_3_F_2_ 8, 10, and 11 populations, in which the BC_3_F_1_ parents did not retain the major QTL regions, might segregate QTLs additively with anther length increases of 0.15, 0.10, and 0.10 mm, respectively, at the SSR markers *RM5755*, *RM3317*, and *RM5635*. The BC_3_F_2_ 25 population did not have the *O. longistaminata* segment at the major QTL regions, but the mean anther length of the population was significantly higher than that of T65 (Figure 3). Two populations—BC_3_F_2_ 6 and 17—also did not have the major QTL region, and so most likely had another, unidentified QTL which could not be detected in the BC_3_F_1_ population. We also inferred chromosomal regions associated with increasing anther length at the major QTLs on Chrs. 3, 5, and 6, but the LOD peak positions were shifted, most likely owing to the few recombinations around QTL regions in populations with only 21 to 24 individuals. This result demonstrates that analysis in BC_3_F_2_ populations covering the whole genome of *O. longistaminata* increased the detection power, and that it was a practical way to perform the QTL scan where whole-genome genotyping is not available, despite reduced accuracy of the inferred map positions. This practical approach will be useful in the design of future experiments. The simple sum of the additive effects of the W1508 alleles at all QTLs reached 2.14 mm when homozygous (Appendix A), which almost explained the difference between W1508 and T65 (4.33 mm − 2.27 mm = 2.06 mm). This suggested that multiple QTLs functioned additively, or by interacting with each other to form anthers as *O. longistaminata*.

The genetic analysis of anther length, which likely reflected on the lifestyle of the perennial wild rice species *O. longistaminata*, revealed that at least four major QTLs, *qATL3.1*, *qATL5*, *qATL6.1*, and *qATL6.2* increased anther length. Regional QTL analysis and CSSL analysis suggested that additional minor QTLs were also associated with regulation of anther length.

## 4. Materials and Methods

### 4.1. Plant Materials

We used *O. longistaminata* accession W1508, which was originally collected in Madagascar. *Oryza sativa* L. ssp. *japonica* Taichung 65 (T65) and W1508 were crossed, to develop F_1_ plants with T65 cytoplasm. The National Institute of Genetics, Mishima, Japan, kindly provided *O. longistaminata* ratton and the seeds of *O. barthii* and *O. glumaepatula* accessions, and F_1_ plants derived from a cross between T65 and W1508. T65 was used as the male parent in recurrent backcrosses to develop BC_3_F_1_ plants (Figure 2). The BC_3_F_2_ population was grown in 2015 and 2016.

### 4.2. Measurement of Anther Length

Panicles were collected at the heading stage and fixed and preserved in 70% ethanol. All anthers from 1 spikelet, just before anthesis, were removed and placed on a glass slide and then photographed under a microscope (Axioplan, Carl Zeiss, Jena, Germany) with a digital camera (DMX-1200, Nikon, Tokyo, Japan). The lengths of 4 to 6 anthers were measured in ImageJ software version 1.8 [38]. The average anther length was used as the phenotypic value of an individual plant.

### 4.3. Genotyping

Adult leaves were freeze-dried (FDU-1200, Eyela, Tokyo, Japan) and ground with a Multi-bead shocker (Yasui Kikai, Osaka, Japan). Genomic DNA was extracted by the potassium acetate method [39] from ground samples. The plants were genotyped for 124 simple sequence repeat (SSR) markers evenly distributed across the rice genomes (Appendix A). For each marker, a 15 μL reaction mixture consisted of 50 mM KCl, 10 mM Tris (pH 9.0), 1.5 mM MgCl_2_, 200 μM dNTPs, 0.2 μM primers, 0.75 U of GoTaq DNA polymerase (Promega, Fitchburg, WI, USA), and ∼10 ng of genomic DNA template. PCR was performed on a GeneAmp PCR System 9700 (Applied Biosystems, Foster City, CA, USA). The thermal profile was an initial denaturation at 95 °C for 5 min: 35 cycles of 95 °C for 30 s, 55 °C for 30 s, and 72 °C for 30 s; and then a final elongation step at 72 °C for 7 min. The amplified products were electrophoresed in 4% agarose gel in 0.5× TBE buffer.

### 4.4. QTL Analysis

Simple interval mapping by an R/qtl library was used to detect QTLs conferring anther length [40]. LOD score thresholds with a significance level of 5% were empirically estimated by 1000 permutation tests. LOD peaks that exceeded the thresholds defined the QTLs. For multiple QTL mapping in the BC_3_F_1_ population, a forward/backward stepwise search was used for model selection, including epistasis of two loci by using the stepwiseqtl function in the R/qtl library, with a penalized LOD score criterion to balance model fitting and model complexity.

### 4.5. Measurement of Numbers of Pollen Grains

Six ethanol-fixed anthers from one spikelet per plant were crushed in a 1.5-mL tube containing 10 μL of 1% I_2_–KI solution, and then suspended in 390 μL of distilled water. One μL of the suspension was applied to a glass slide, and pollen grains in 10 fields were photographed under a light microscope (Axioplan, Carl Zeiss, Jena, Germany). The pollen grains were counted in each field, and the numbers were multiplied by the dilution ratio to estimate the pollen grain number per anther.

### 4.6. Observation of Epidermal Cells of Anthers

Anthers were put in a drop of distilled water on a glass slide, and cut at both ends, and pollen was removed by pipetting back and forth. The epidermal cells of three anthers from one spikelet per plant were photographed at the center in the long axis of the anthers, through an optical microscope (Axioplan, Carl Zeiss, Jena, Germany) with a digital camera (DMX-1200, Nikon, Tokyo, Japan), and were counted in ImageJ. Three plants per line were evaluated as a replicate each, and the average value of epidermal cell length of nine anthers was calculated.

## 5. Conclusions

Genetic analysis of anther length, which is likely to relate to an outcrossing lifestyle in the perennial wild rice species *O. longistaminata*, revealed that at least four major QTLs, *qATL3*, *qATL5*, *qATL6.1*, and *qATL6.2* increase anther length. The regional QTL analysis and constructed CSSL series (designated ‘W1508IL’) suggested additional minor QTLs associated with the regulation of anther length. The cloning and diversity analysis of genes conferring anther length QTLs promotes utilization of the genetic resources of wild species, and the understanding of haplotype evolution on the differentiation of annuality and perenniality in the genus *Oryza*.

## Figures and Tables

**Figure 1 plants-08-00388-f001:**
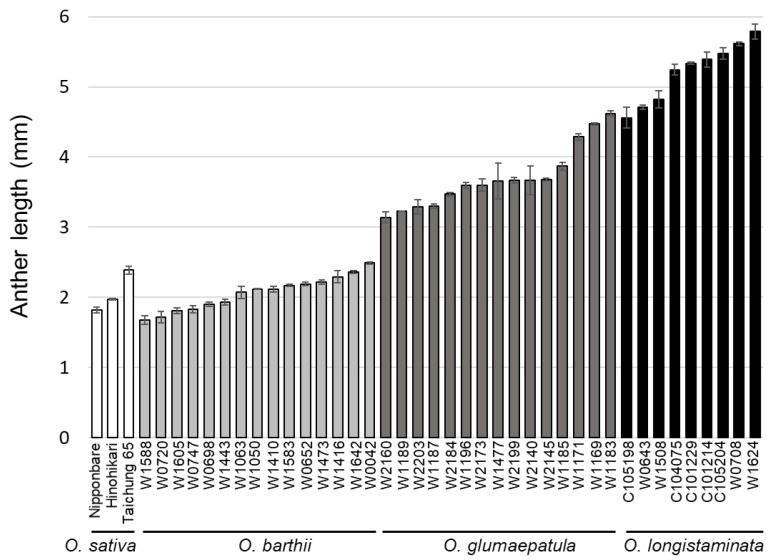
Distribution of anther length (mm) in cultivars of *O. sativa* and accessions of *O. barthii*, *O. glumaepatula*, and *O. longistaminata*. Mean ± SE, n = 3.

**Figure 2 plants-08-00388-f002:**
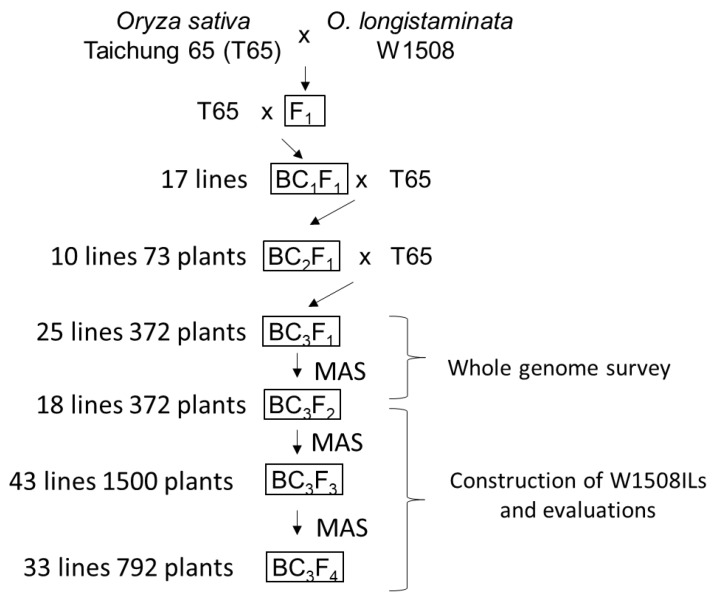
Breeding scheme of genetic materials used in this study. MAS represents marker-assisted selection.

**Figure 3 plants-08-00388-f003:**
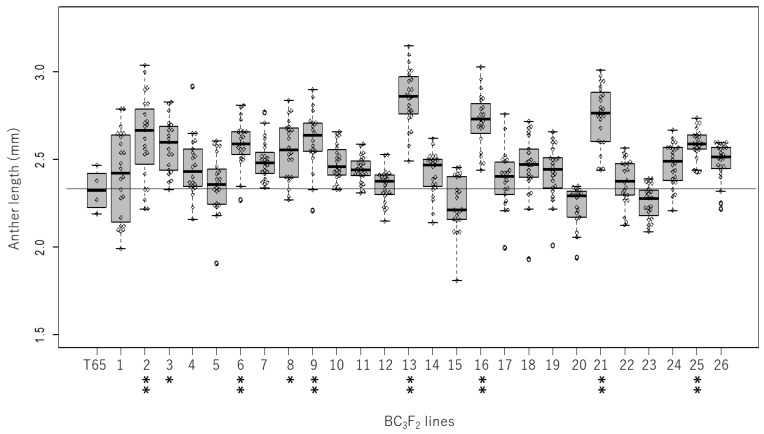
Box plots of anther length among the 26 BC_3_F_2_ populations. A thin horizontal line shows the average anther length in the Taichung 65 (T65) genetic background. Thick horizontal bars represent average values. Ranges between upper and lower quantiles are indicated by grey boxes. Maximum and minimum values excluding outlier are indicated by upper and lower whiskers, respectively. * and ** represent significant differences at P = 0.05 and P = 0.01, respectively.

**Figure 4 plants-08-00388-f004:**
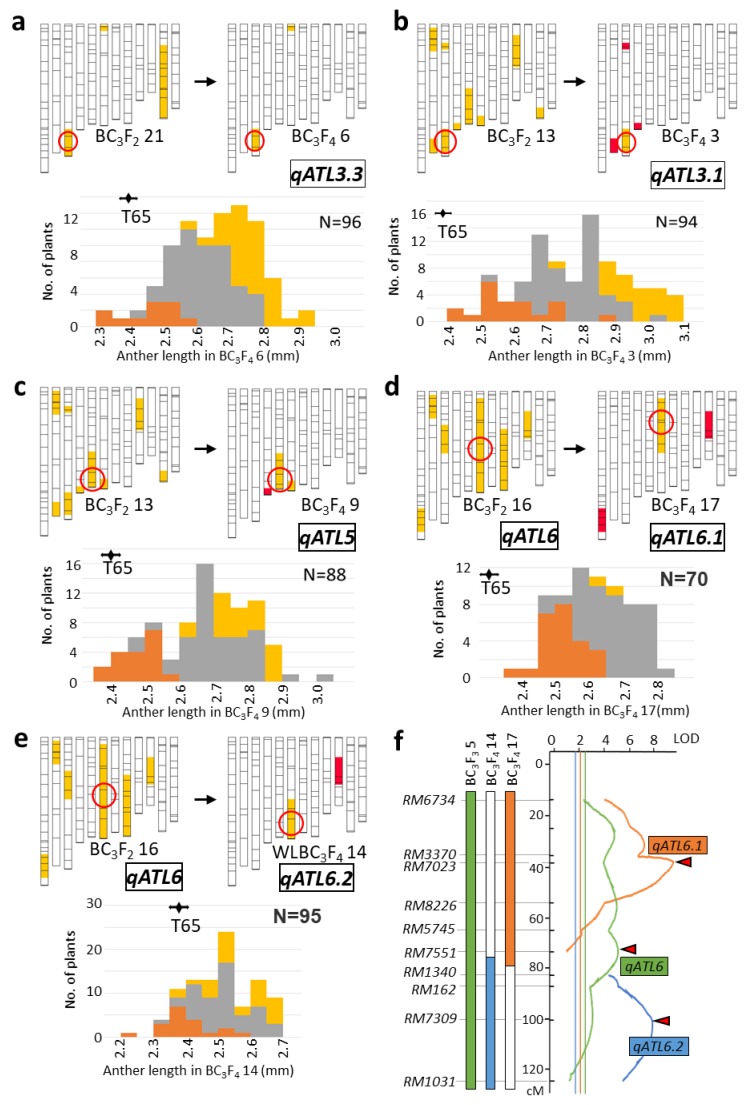
Validation of the anther length quantitative trait locus (QTLs) located on chromosomes 3, 5, and 6 in BC_3_F_4_ populations segregating at only one QTL region. (**a**)–(**e**) Graphical genotypes of the parents of BC_3_F_3_ and BC_3_F_4_ populations, and frequency distributions of anther length in BC_3_F_4_ populations for (**a**) *qATL3.3*, (**b**) *qATL3.1*, (**c**) *qATL5*, (**d**) *qATL6.1*, and (**e**) *qATL6.2*. Graphical genotypes: Yellow, heterozygous; red, homozygous for the W1508 allele. Frequency distributions shown in colors by genotypes at *RM5959* (**a**), *RM3525* (**b**), *RM1054* (**c**), *RM7023* (**d**), and *RM7309* (**e**): Orange, homozygous for T65; yellow, homozygous for W1508; gray, heterozygous; (**f**) logarithm of odds (LOD) curves in simple interval mapping of anther length for detection of *qATL6* (green) in BC_3_F_3_ 5, *qATL6.1* (orange) in BC_3_F_4_ 17, and *qATL6.2* (blue) in BC_3_F_4_ 14.

**Figure 5 plants-08-00388-f005:**
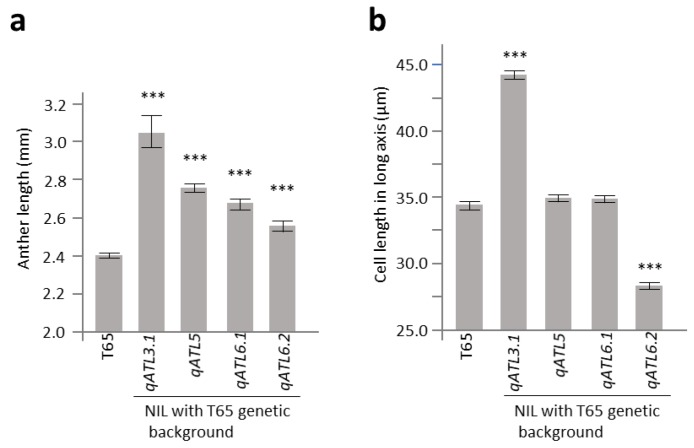
Characterization of QTLs using nearly isogenic lines (NIL) carrying W1508 chromosomal segments around QTL regions detected in this study. (**a**) Anther length (mm) in the NILs. Mean ± SE, n = 3; (**b**) cell length (µm) in the long axis of anthers in the NILs. * and ** represent significant differences in Dunnett’s multiple comparisons to T65 control at 5% and 1% levels, respectively. The numbers of observed cells were 1460, 997, 953, 879, and 1033 in *qATL3.1* NIL, *qATL5* NIL, *qATL6.1* NIL, *qATL6.2* NIL, and T65 respectively.

**Figure 6 plants-08-00388-f006:**
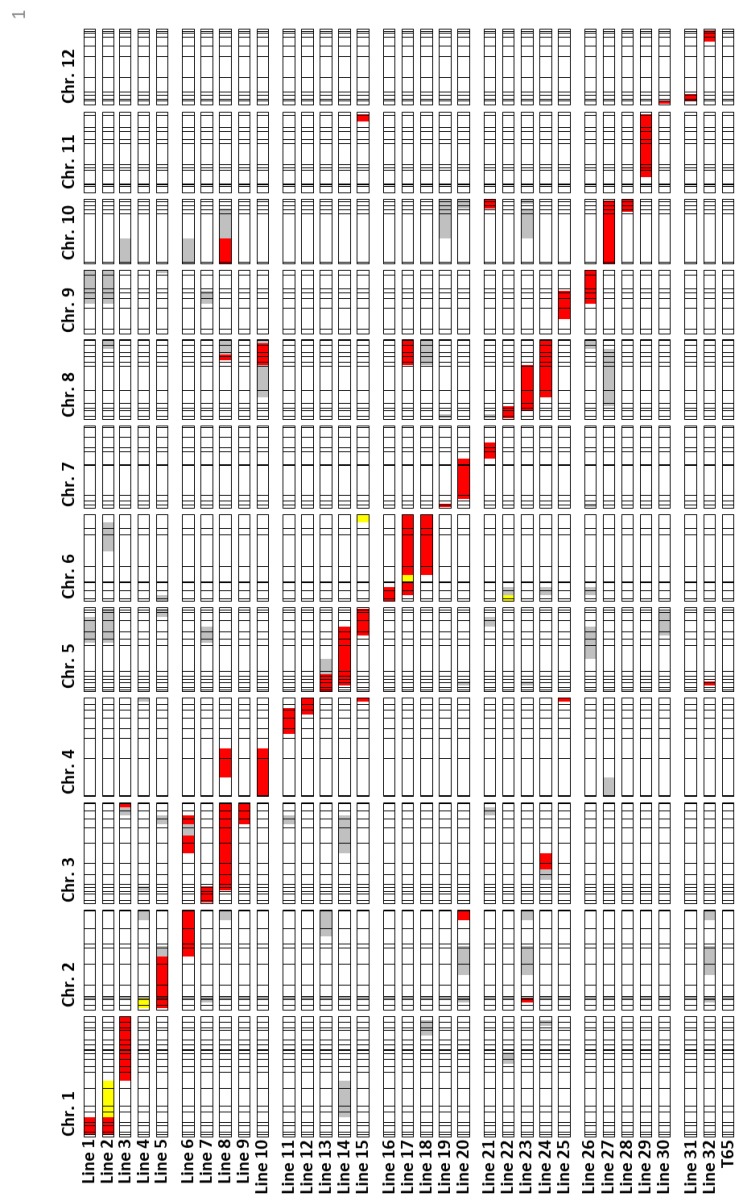
Construction of chromosome segment substitution lines of *O. longistaminata* accession W1508 (W1508ILs) in the genetic background of *O. sativa* L. cv. Taichung 65. Red and white boxes represent homozygous genotypes for W1508 and Taichung 65 alleles, respectively. Yellow boxes represent heterozygous genotypes. Missing genotypes at markers showing heterozygous genotypes at BC_3_F_1_ generation are indicated by grey.

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
