# Peer review of "Identification of Anther Length QTL and Construction of Chromosome Segment Substitution Lines of Oryza longistaminata"

_plants, 2019, doi:10.3390/plants8100388_

Round 1

Reviewer 1 Report

I suggest a title like - Identification of QTL regulating anther length in Oryza longistaminata

Abstract

Need to provide the size of mapping populations and number and type of markers used for linkage map and QTL analysis.

“In the genome-wide QTL analysis of “ – “QTL analysis of..”

“Thirteen possible QTLs” – This is confusing

Introduction

“a by using advanced backcrossed QTL analysis” – inappropriate

Results

Need to use the abbreviation IL, and CSSLs consistently.

Before starting with QTLs results, the number of progenies/lines used to genotype and phenotypes need to make very clear. The population generated and used for QTL mapping looks very confusing. Make it simpler and straight forward. For instance “the 26 BC3F2 populations” has how many progenies that used for genotyping and phenotyping.

Chrs. 1–6 and 9–11 – not clear the nomenclature of Chrs.

I suggest not to abbreviate Anther Length.

Overall MS looks excellent. 

Author Response

We deeply appreciate your careful reviewing.
The following is our revision responded to the comments from the Reviewer 1.

[Comment 1]
Abstract
Need to provide the size of mapping populations and number and type of markers used for linkage map and QTL analysis.
The size of mapping populations, number and types of markers were provided in the text.
[Response to the Comment 1]
We added the size of mapping populations and number and type of markers used for genetic analysis in in Abstract section.
In body text, we added marker type (SSR markers) in the Materials and Methods section.
For size of mapping populations, please see L97, L123, Table S1, and the subsection "2.2 Validation of QTL".

[Comment 2]
“In the genome-wide QTL analysis of “ – “QTL analysis of..”
[Response to the Comment 2]
We revised in the new manuscript because "QTL analysis of" becomes a subject of this sentence
according to the reviewer's comment.

[Comment 3]
“Thirteen possible QTLs” – This is confusing
[Response to the Comment 3]
Thank you for your comment.
Here we aimed to show that minor QTLs also were possible to be suggested using BC3F2 population and were motivated to develop chromosome segment substitution lines for consequent identification of minor QTLs. Therefore, what we want to mention here is not exact numbers of QTLs but the minor QTLs also were possible to be suggested in BC3F2.
According to this concept, we revised the corresponding sentence as below.
(old)
Thirteen possible QTLs were identified in the regional QTL analysis in BC3F2 populations
=>
(new)
The additional minor QTLs were suggested in the regional QTL analysis using BC3F2 populations.
Accompany with this revision in the abstract, we revised the body text at L132.

[Comment 4]
Introduction
“a by using advanced backcrossed QTL analysis” – inappropriate
[Response to the Comment 4]
Thank you for your careful reading.
We revised as follows in the whole manuscript.
(old)
advanced backcrossed QTL
=>
(new)
by using advanced backcross QTL

[Comment 5]
Results
Need to use the abbreviation IL, and CSSLs consistently.
[Response to the Comment 5]
Thank you for your careful reading.
In the revised manuscript, we used "CSSLs" to indicate general meaning of chromosome segment substitution lines
whereas we used "ILS" as a part of name of W1508ILs (CSSLs derived from W1508) .
We consider we have remove confused usage of the terms for consistency in the revised manuscript.

[Comment 6]
Before starting with QTLs results, the number of progenies/lines used to genotype and phenotypes need to make very clear. The population generated and used for QTL mapping looks very confusing. Make it simpler and straight forward. For instance “the 26 BC3F2 populations” has how many progenies that used for genotyping and phenotyping.
[Response to the Comment 6]
We deeply apologize reviewers' confusions and consumption of valuable time for reading.

[Comment 7]
Chrs. 1–6 and 9–11 – not clear the nomenclature of Chrs.
[Response to the Comment 7]
Thank you for your comment. We revised as follows.
(old)
Chrs. 1–6 and 9–11
=>
(new)
Chrs. 1, 2, 3, 4, 5, 6, 9, 10, and 11.

[Comment 8]
I suggest not to abbreviate Anther Length.
[Response to the Comment 8]
Thank you for your suggestion.
We spell out anther length over whole manuscript.

Reviewer 2 Report

Abstract:
Neither in the abstract nor in the conclusion, you mentioned some perspectives from your results.

Introduction:
- l 30-35: "Annual 30 species allocate their resources primarily to sexual reproduction, whereas perennial species allocate their resources primarily to vegetative growth, depending on the ecological circumstances. Annuals tend to produce as many flowers as possible to increase the one-time dispersal of seeds. By contrast, perennial species tend to allocate resources to vegetative growth because of the need to occupy physical space and use local water and nutrient resources over an extended lifetime [1]": There is in these two sentences some redundancy which should be amended.
- l 35-37: "As a result, perennials might have higher genetic potential to adapt to various biotic and abiotic stresses due to high heterozygosity": I do not understand why you wrote "as a result"; heterozygosity might be involved in the process, but you did not bring any evidence or reference for that in the previous statements.
- l 41-42: ".. contribute to the perenniality and their outcrossing characteristics?": this study gives insights in phenotypes involved in outcrossing characteristics, but not at all in perenniality. At lines 55-62, you mentioned some apparent correlation between perenniality and outcrossing/heterozygosity, but either this seems to be highly speculative, or you did not provide, from previous works, evidence for this.

Results:
- l 94-95: "...all with the effect of increasing AL": this is not accurate enough. Would you mean that O.longistamina brought all the QTL alleles increasing the anther length ?
- l 103-104: "Next, we aimed to seek as many QTLs as possible within the limited genotyping capacities of the study": this sentence is not useful, and does not justify your (right) choice to develop NIL's.
- l 124: "One QTL positively regulating AL": "modulating" rather than "regulating" ?

Discussion:
- l 235-240: It is not clear whether the experimental design and/or the algorithm used to perform the QTL analysis (see 5.4 QTL analysis) were able to detect epistatic effects between qATL3.1, qATL3.3, qATL5, qATL6.1 and qATL6.2.

Author Response

I deeply appreciate the critical and careful reading for logical writing of the manuscript by the reviewers.
Please check the revised manuscript as to satisfy the reviewers' comments.

[Comment 1]
Abstract:
Neither in the abstract nor in the conclusion, you mentioned some perspectives from your results.
[Response to the Comment 1]
Thank you for giving us oppotunity to write future perspectives. We added the sentence below.
Please see the abstract and conclusion sections.

The cloning and diversity analysis of gene conferring anther length QTL promote utilization of genetic resources of wild species and understanding haplotype evolution on differentiation of annuality and perenniality in genus Oryza.

[Comment 2]
Introduction:
- l 30-35: "Annual 30 species allocate their resources primarily to sexual reproduction, whereas perennial species allocate their resources primarily to vegetative growth, depending on the ecological circumstances. Annuals tend to produce as many flowers as possible to increase the one-time dispersal of seeds. By contrast, perennial species tend to allocate resources to vegetative growth because of the need to occupy physical space and use local water and nutrient resources over an extended lifetime ": There is in these two sentences some redundancy which should be amended.
[Response to the Comment 2]
Thank you for your comment.
We revised the manuscript to remove redundancy as follow.

Annuals tend to allocate their resources to sexual reproduction to produce as many flowers as possible for the one-time dispersal of seeds. By contrast, perennial species tend to primarily allocate resources to vegetative growth because of the need to occupy physical space and use local water and nutrient resources over an extended lifetime depending on the ecological circumstances [1]

[Comment 3]
- l 35-37: "As a result, perennials might have higher genetic potential to adapt to various biotic and abiotic stresses due to high heterozygosity": I do not understand why you wrote "as a result"; heterozygosity might be involved in the process, but you did not bring any evidence or reference for that in the previous statements.
[Response to the Comment 3]
Thank you for your comment. The tendency of higher heterozygosity in perennial species is a results of observation (fact). We should not say "As a result".
We removed "as a result", and cited the references to represent heterozygosity in perennial species showed correlation to fittness-related traits such as survival probability, reproductive success, and disease resistance. Please see L38-L40 in the revised manuscript.

[Comment 4]
- l 41-42: ".. contribute to the perenniality and their outcrossing characteristics?": this study gives insights in phenotypes involved in outcrossing characteristics, but not at all in perenniality. At lines 55-62, you mentioned some apparent correlation between perenniality and outcrossing/heterozygosity, but either this seems to be highly speculative, or you did not provide, from previous works, evidence for this.
[Response to the Comment 4]
As the reviewer pointed out, we also consider that outcrossing-related traits are neither nececcesary condition nor sufficient conditions. We omitted perenniality from our statement as products of this study.
Please check L44-46.

[Comment 5]
Results:
- l 94-95: "...all with the effect of increasing AL": this is not accurate enough. Would you mean that O.longistamina brought all the QTL alleles increasing the anther length ?
[Response to the Comment 5]
Thank you for your kind correction. Our explanation is not accurate enough.
As the reviewer's comment, we revised as follow.
(old)
Simple interval mapping suggested one QTL on Chr. 3, one QTL on Chr. 5, and two QTLs on Chr. 6, all with the effect of increasing AL (Table S1).
=>
(new)
Simple interval mapping suggested that O. longistamina alleles at one QTL on Chr. 3, one QTL on Chr. 5, and two QTLs on Chr. 6 increased the anther length.

[Comment 6]
- l 103-104: "Next, we aimed to seek as many QTLs as possible within the limited genotyping capacities of the study": this sentence is not useful, and does not justify your (right) choice to develop NIL's.
[Response to the Comment 6]
Thank you for your suggestion. We omitted this sentence.

[Comment 7]
- l 124: "One QTL positively regulating AL": "modulating" rather than "regulating" ?
[Response to the Comment 7]
Thank you for your suggestion.
"Regulate" or "control" has been more widely used to explain phenotypic variations were changed by genotypes of (Mendelian) genes/QTLs because ,in my opinion, gene (or genotypes) is a "determinant" of genetic values in statistical genetics. On the other hand, modulate include feeling "adjust" physiological level of target by feedback system.

Here we found the QTL (gene) qATL3.3 additively increasing genetic values of anther length. This QTL
is suggested to determine additive and dominance effect without dependency of genotypes at other loci,
this qATL3.3 is determinative to increase anther length.

Therefore we believe "regulating" is better in present situation.
Of course we did not decline possibility that qATL3.3 have a modulator function of some kinds of
protein or transcript level during anther development in future gene cloning.

[Comment 8]
Discussion:
- l 235-240: It is not clear whether the experimental design and/or the algorithm used to perform the QTL analysis (see 5.4 QTL analysis) were able to detect epistatic effects between qATL3.1, qATL3.3, qATL5, qATL6.1 and qATL6.2.
[Response to the Comment 8]
Thank you for your critical comment. We did not care of apparent mention in Materials and Methods and body text.
At L99 ~ L101 on the Result section, we revised as follow.

(old)
Multiple QTL analysis using forward/backward model selection also identified these four QTLs as explaining more than half (55.5%) of the phenotypic variation of AL in the BC3F1 population (Figure S2, Table S1).
=>
(New)
Multiple QTL analysis using forward/backward model selection also suggested that these four QTLs additively increase anther length without epistasis as explaining more than half (55.5%) of the phenotypic variation in the BC3F1 population (Figure S2, Table S1).

At L316 ~ L318 on the Materials and Methods section (5.4 QTL analysis), we revised as follow.
(old)
For multiple QTL mapping in the BC3F1 population, a forward/backward stepwise search was used for model selection by using the stepwiseqtl() function in R/qtl library with a penalized LOD score criterion to balance model fitting and model complexity.
=>
(New)
For multiple QTL mapping in the BC3F1 population, a forward/backward stepwise search was used for model selection including epistasis of two loci by using the stepwiseqtl() function in R/qtl library with a penalized LOD score criterion to balance model fitting and model complexity.

Reviewer 3 Report

Authors revealed several QTLs explaining the different anther length between Oryza sativa, T65 and O. longistaminata, W1508 and these findings are scientifically valuable and should be published in this journal.

qATL3.1 and qATL3.3 were both detected on long arm of chromosome 3 and both position were estimated as 142.5 cM and 151.5 cM respectively. Both QTLs were validated in different BC3F4 populations (BC3F4 3 and BC3F4 6) having the same graphical genotype at long arm of chromosome 3. Thus, when author name two QTLs on chromosome 3, authors should  show supporting data, which distinguish qATL3.1 from qATL3.3, such as , QTL LOD curves using additional DNA markers between 142.5 cM and 151.5 cM and anther length data of NIL which contains W1508 chromosome segment on the QTL region.

While authors can treat the QTL located on chromosome 3  as one QTL and  refer to the possibility that they are two different QTLs in the region.

Author Response

We deeply appreciate your careful reviewing.
The following is our revision responded to the comments from the Reviewer 3.

[Comment 1]
qATL3.1 and qATL3.3 were both detected on long arm of chromosome 3 and both position were estimated as 142.5 cM and 151.5 cM respectively. Both QTLs were validated in different BC3F4 populations (BC3F4 3 and BC3F4 6) having the same graphical genotype at long arm of chromosome 3. Thus, when author name two QTLs on chromosome 3, authors should show supporting data, which distinguish qATL3.1 from qATL3.3, such as , QTL LOD curves using additional DNA markers between 142.5 cM and 151.5 cM and anther length data of NIL which contains W1508 chromosome segment on the QTL region.

While authors can treat the QTL located on chromosome 3 as one QTL and refer to the possibility that they are two different QTLs in the region.

[Response to the Comment 1]
Thank you for your careful reading.
We tentatively handled two qATL3.1 and qATL3.3 as a different QTLs because they were detected using different populations, but we would like to conclude that they are identical QTLs on chromosome 3, designated qATL3.1.

We added the following sentence at L169 - L171.
qATL3.3 and qATL3.1 seemed to be identical QTLs because substitution of O. longistaminata chromosome segment raged from the SSR markers RM3525 and RM5959. We unified the QTL names qATL3.3 and qATL3.1 as the single QTL qATL3.1.

Reviewer 4 Report

This paper reports a genetic dissection by GWAS of the genetic control of anther length in a wild rice species relative to cultivated common rice. Anther length (and size) is presumably related to an outcrossing lifestyle but could be beneficial for producing hybrids of cultivated rice. Accession W1508 of O. longistaminatawas crossed and backcrossed to common rice cultivar Taichung 65 and introgressed chromosome regions of the donor parent were identified and fixed in the recurrent parent by means of genetic markers. QTL analyses of anther length was performed on of some of the segregating materials. The genome of W1508 was largely recovered in a series of 32 introgression lines (Figure 6). Four QTL located on three chromosomes accounted for much of the variation in anther length between W1508 and T65 but there was evidence that additional less effective QTL were also involved.

Overall, the paper is well presented. However, the English expression could be improved. A copy of the manuscript PDF with pencilled suggestions is attached.

Author Response

Thank you for careful reviewing.
We would like to perform GWAS of anther length in O. longistaminata with a lot of interests, but we could not include the result of them. In this manuscript, we have shown the results of QTL  analysis as the reviewer pointed out.

We completely follow the English editing by the reviewer 4 and revised the manuscript.

Figures 4 and 5.
The reviewer pointed out to keep a consistency in style of alphabet representing each panel of figures 4 and 5.
We had use bold alphabets (without parenthesis) for heading panel names like a-e to indicate multiple panels in figure 4 and (a), (b), (c), (d), and (e) styles to indicate panels in the sentence. Therefore we believe we had a consistency for the style.
Therefore we think we do not need to revise here.
